# Cabozantinib Sensitizes NSCLC Cells to Radiation by Inducing Ferroptosis via STAT3/MCL1/BECN1/SLC7A11 Axis Suppression

**DOI:** 10.3390/cancers17182950

**Published:** 2025-09-09

**Authors:** Cheng-Yi Wang, Chao-Yuan Huang, Li-Ju Chen, Grace Chen, Shiao-Ya Hong

**Affiliations:** 1Department of Internal Medicine, Cardinal Tien Hospital and School of Medicine, College of Medicine, Fu Jen Catholic University, New Taipei City 231009, Taiwan; cywang@mospital.com; 2Division of Radiation Oncology, Department of Oncology, National Taiwan University Hospital, College of Medicine, National Taiwan University, Taipei 100229, Taiwan; cyhuang999@ntu.edu.tw (C.-Y.H.); lijuchen16@mospital.com (L.-J.C.); 3School of Medicine, College of Medicine, Taipei Medical University, Taipei 110301, Taiwan; b101110127@tmu.edu.tw; 4Department of Biotechnology and Laboratory Science in Medicine, National Yang Ming Chiao Tung University, Taipei 112304, Taiwan

**Keywords:** NSCLC, radiotherapy, ferroptosis, cabozantinib, MCL1, STAT3, BECN1, SLC7A11

## Abstract

This study investigates how cabozantinib, a clinically approved multi-kinase inhibitor (TKI), sensitizes non-small-cell lung cancer (NSCLC) cells to radiation by inducing ferroptosis, a form of lipid peroxidation-driven cell death. Among the three VEGFR-targeting tyrosine kinase inhibitors tested, only cabozantinib enhanced radiation-induced cytotoxicity through ferroptosis, as shown by lipid ROS accumulation, glutathione depletion, and reversal with ferroptosis inhibitors. Mechanistically, cabozantinib inhibited STAT3 and MCL1, releasing BECN1 to suppress SLC7A11 and disrupt redox balance. Silencing BECN1 abrogated these effects, confirming its central role. Importantly, structurally related TKIs (lenvatinib, ripretinib) lacked these actions, suggesting that the inhibition of non-VEGFR targets (e.g., MET, AXL) is critical. This work identifies a previously unrecognized mechanism by which cabozantinib triggers ferroptotic radiosensitization in NSCLC cells via the STAT3/MCL1/BECN1/SLC7A11 axis. These findings highlight ferroptosis as a therapeutically targetable vulnerability in redox-adapted, radiation-resistant tumors and propose a novel strategy to improve radiotherapy outcomes using multi-target kinase inhibitors.

## 1. Introduction

Radiotherapy is a standard-of-care modality for non-small-cell lung cancer (NSCLC), particularly in locally advanced or unresectable cases [1,2]. However, its curative efficacy is often undermined by intrinsic resistance mechanisms that allow tumor cells to adapt to and withstand oxidative stress. A major contributor to this resistance is the ability of cancer cells to buffer radiation-induced reactive oxygen species (ROS) through tightly regulated redox homeostasis, which in turn prevents the execution of oxidative cell death programs [3,4]. This presents a critical barrier to radiation efficacy that remains incompletely addressed in current clinical practice.

Among emerging modes of non-apoptotic cell death, ferroptosis has gained attention as a promising radiosensitization strategy due to its dependence on lipid ROS accumulation and its vulnerability to redox imbalance [5,6]. The execution of ferroptosis is tightly controlled by antioxidant systems such as glutathione peroxidase 4 (GPX4) and system Xc^−^, which imports cystine via SLC7A11 to support glutathione synthesis [7,8,9]. The elevated expression of these components confers ferroptosis resistance and has been implicated in poor response to ROS-inducing therapies [10]. While pharmacologic inducers of ferroptosis (e.g., erastin, RSL3) have shown preclinical efficacy, their systemic toxicity and poor tumor specificity limit translational potential [11], emphasizing the need to identify clinically viable modulators of ferroptosis pathways.

Beclin 1 (BECN1) has recently identified as a regulator of ferroptosis that is mechanistically distinct from its canonical role in autophagy. While autophagy requires BECN1 to assemble the class III PI3K complex, ferroptosis induction involves its direct interaction with SLC7A11 [12,13]. By binding to SLC7A11, BECN1 inhibits transporter activity without altering its expression, thereby limiting cystine uptake, depleting glutathione, and promoting lipid peroxidation [13]. This inhibitory interaction is reinforced by the AMP-activated protein kinase (AMPK)-mediated phosphorylation of BECN1 at Ser90/93/96, while the disruption of this phosphorylation abrogates ferroptosis induction [13]. Importantly, the availability of BECN1 to engage SLC7A11 is tightly controlled by oncogenic survival pathways. In NSCLC and other malignancies, persistent STAT3 activation upregulates its downstream effector MCL1 (a member of the BCL-2 family), which sequesters BECN1 away from SLC7A11 and thereby preserves system Xc^−^ function and ferroptosis resistance [14,15]. In this context, STAT3-driven MCL1 expression acts as a molecular brake, maintaining cystine import and redox homeostasis. Conversely, the suppression of STAT3/MCL1 signaling releases BECN1 to bind SLC7A11, shifting the balance toward ferroptotic cell death. Thus, the STAT3/MCL1/BECN1/SLC7A11 axis represents a key regulatory hub that couples upstream survival signaling to the execution of ferroptosis.

Current evidence indicates that several tyrosine kinase inhibitors (TKIs) can induce ferroptosis. Cabozantinib is a clinically approved multi-target TKI with inhibitory activity against VEGFRs, MET, AXL, and RET [16]. While its therapeutic utility has primarily been attributed to anti-angiogenic and oncogenic signaling suppression, its effects on redox-regulating pathways remain poorly defined. Given its broad kinase profile, we hypothesized that cabozantinib may interfere with STAT3-driven redox defense and promote ferroptosis, thereby enhancing the efficacy of radiotherapy in NSCLC. To test this, we systematically evaluated whether cabozantinib could modulate ferroptosis execution and radiosensitivity in NSCLC cell models, with a focus on the STAT3/MCL1/BECN1/SLC7A11 axis as a candidate mechanism.

## 2. Materials and Methods

### 2.1. Cell Culture and Treatments

Human NSCLC cell lines H1299 and A549 were obtained from the American Type Culture Collection (ATCC, Manassas, VA, USA). Cells were maintained in RPMI-1640 medium supplemented with 10% fetal bovine serum (FBS), 2 mM L-glutamine, 10 mM HEPES, 2000 mg/L glucose, 1 mM sodium pyruvate, and 1500 mg/L sodium bicarbonate. All cultures were incubated at 37 °C in a humidified atmosphere containing 5% CO_2_. Mycoplasma contamination was routinely monitored using the EZPCR Mycoplasma Test Kit (Biological Industries, Beit HaEmek, Israel). Cells were treated with the indicated concentrations of cabozantinib, lenvatinib, or ripretinib (MedChemExpress, USA) for 24 h. Irradiation was performed using an IBL637 gamma irradiator (CIS Bio International, Saclay, France) delivering 4 Gy of Caesium-137 gamma rays at a dose rate of 3.373 Gy/min at room temperature. For combination treatments, cells were pretreated with cabozantinib followed by irradiation. In inhibitor experiments, cells were pretreated with 5 μM ferrostatin-1 (Fer-1) or 200 μM deferoxamine (DFO) for 1 h prior to cabozantinib exposure. 

### 2.2. Cell Viability Assay

The 3-[4,5-dimethylthiazol-2-yl]-2,5-diphenyl-tetrazolium bromide (MTT) assay was performed to detect cell viability. Briefly, A549 cells were seeded in 96-well plates at a density of 3 × 10^3^ cells per well and allowed to adhere overnight. Following treatment, MTT reagent (Sigma-Aldrich, St. Louis, MO, USA) was added to each well and incubated for 3 h at 37 °C. The medium was then carefully removed, and the resulting formazan crystals were solubilized in dimethyl sulfoxide (DMSO). Absorbance was measured at 570 nm using a microplate reader to assess cell viability.

### 2.3. Immunoblotting Analysis

Cell lysates were prepared in RIPA buffer (50 mM Tris-HCl, pH 7.4, 150 mM NaCl, 0.1% SDS, 1% Triton X-100, 0.5% sodium deoxycholate, 2 mM EDTA) supplemented with protease and phosphatase inhibitors (Roche), and total protein concentrations were determined using the BCA assay (Thermo Fisher Scientific, Waltham, MA, USA). Equal amounts of proteins were separated by SDS-PAGE and transferred onto PVDF membranes. Membranes were incubated with primary antibodies, followed by HRP-conjugated secondary antibodies. Protein bands were visualized using enhanced chemiluminescence (ECL; Millipore, Burlington, MA, USA) and quantified using ImageJ software version 1.53k. Primary antibodies were obtained from Cell Signaling Technology and included phospho-STAT3 (Tyr705) (#9145), total STAT3 (#9139), MCL1 (#4572), BECN1 (#4122), SLC7A11 (#12691), and GAPDH (#5174).

### 2.4. Lipid Peroxidation Assays

Lipid ROS levels were assessed using C11-BODIPY (581/591) dye (Thermo Fisher Scientific). Cells were seeded in 6-well plates at a density of 3 × 10^5^ cells per well and treated with the indicated drugs and/or radiation. After treatment, cells were harvested by trypsinization, washed with PBS, and incubated with 2.5 μM C11-BODIPY at 37 °C for 20 min in the dark. Stained cells were then resuspended in Hank’s Balanced Salt Solution (HBSS) and analyzed by flow cytometry using a CytoFLEX flow cytometer (Beckman Coulter, Brea, CA, USA), with a minimum of 10,000 cells collected per sample. The oxidation-induced fluorescence shift from red (590 nm) to green (510 nm) was quantified using CytExpert 2.4 software, and the results are expressed as mean fluorescence intensity (MFI). Intracellular lipid peroxidation was assessed by detecting malondialdehyde (MDA)-modified proteins using a fluorescein-conjugated MDA-specific antibody. Cells were washed with PBS and fixed with 4% paraformaldehyde for 15 min at room temperature. After permeabilization with 0.1% Triton X-100 in PBS for 10 min, cells were blocked with 5% BSA for 1 h. Subsequently, cells were incubated with MDA-FITC monoclonal antibody (Abcam, ab27615) at a 1:100 dilution in blocking buffer for 1 h at room temperature in the dark. After washing, fluorescence signals were detected using a fluorescence microscope or quantified by flow cytometry (excitation/emission: 495/519 nm). MFI was analyzed to determine relative levels of lipid peroxidation.

### 2.5. PI Staining Assay

Following treatment, cells were harvested and washed twice with ice-cold phosphate-buffered saline (PBS). Cells were then incubated with a propidium iodide (PI) solution (2 μg/mL in PBS) for 15 min at 37 °C in the dark. After staining, cells were washed with 0.2% PBST (PBS with 0.2% Tween-20) and resuspended in 500 μL of PBS. Flow cytometric analysis was performed within 1 h using a CytoFLEX flow cytometer. The percentage of PI-positive cells was used to quantify plasma membrane integrity loss, indicative of late-stage cell death.

### 2.6. Colony and Sphere Formation Assays

Colony formation assays were conducted by seeding 500 cells per well into 24-well plates following treatment. After 10 to 14 days of incubation, colonies were fixed with 4% paraformaldehyde for 15 min, stained with crystal violet, and washed. For quantification, the stained colonies were solubilized in methanol, and absorbance was measured at 570 nm (OD570) using a plate reader. Sphere formation assays were performed by plating 5000 cells per well into 6-well Ultra-Low Attachment plates (Corning, NY, USA) in serum-free RPMI-1640 medium supplemented with 10 ng/mL human EGF, 10 ng/mL bFGF, and B27 supplement (Gibco). Cells were cultured under non-adherent conditions for 10 to 14 days, and spheroids larger than 50 μm were imaged and counted using a microscope.

### 2.7. Glutamate Release and Glutathione Assays

Extracellular glutamate release was measured using the Glutamate-Glo™ Assay Kit (Promega), following the manufacturer’s protocol. Total intracellular glutathione (GSH) levels were quantified using the Glutathione Assay Kit (Abcam), suitable for cell and tissue lysates. All assay results were normalized to total protein content, determined by the bicinchoninic acid (BCA) assay (Thermo Fisher). This normalization controlled for variations in cell number, and the data are expressed as a percentage relative to the control group.

### 2.8. Co-Immunoprecipitation

Cells were then washed with cold PBS and lysed in ice-cold IP lysis buffer (50 mM Tris-HCl, pH 7.4, 150 mM NaCl, 2 mM EDTA, 0.1 % NP40) supplemented with protease and phosphatase inhibitors. Lysates were incubated on ice for 30 min and collected by centrifugation at 13,000× *g* for 15 min at 4 °C. For each sample, 1000 μg of total protein was incubated overnight at 4 °C with 1 μg of anti-BECN1 antibody (Cell Signaling Technology, #3495). Immune complexes were captured by adding 25 μL of Protein G Sepharose 4 Fast Flow beads (Sigma-Aldrich, #17061801) and incubating for an additional 1 h at 4 °C with rotation. Beads were washed four times with ice-cold lysis buffer and resuspended in a 5× Laemmli sample buffer. Samples were boiled for 5 min and subjected to SDS-PAGE and immunoblotting. Input lysates (15 μg) were included as loading controls.

### 2.9. BECN1 Knockdown by siRNA and shRNA

For the transient silencing of BECN1, cells were transfected with ON-TARGETplus Human BECN1 (8678) siRNA (L-010552-00-0005, target sequence: 5′-GAUACCGACUUGUUCCUUA-3′, 5′-GGAACUCACAGCUCCAUUA-3′, 5′-CUAAGGAGCUGCCGUU-AUA-3′, 5′-GAGAGGAGCCAUUUAUUGA-3′; Dharmacon, Lafayette, CO, USA) or the corresponding ON-TARGETplus Non-targeting Pool (D-001810-10-20, target sequence: 5′-UGGUUUACAUGUCGACUAA-3′, 5′-UGGUUUACAUGUUGUGUGA-3′, 5′-UGGUUU-ACAUGUUUUCUGA-3′, 5′-UGGUUUACAUGUUUUCCUA-3′; Dharmacon) as a negative control using the DharmaFECT 4 transfection reagent, Dharmacon (Lafayette, CO, USA), according to the manufacturer’s instructions. For stable knockdown, cells were transduced with lentiviral vectors encoding BECN1 shRNA (TRCN299864, target sequence: 5′-CCAGATGCGTTATGAAAAGTA-3′; National RNAi Core Facility, Academia Sinica, Taipei, Taiwan) or a scrambled shRNA construct as the control and then selected with puromycin (5 μg/mL) for 4 days post-transduction. Knockdown efficiency was validated by immunoblotting for BECN1 protein expression prior to functional assays.

### 2.10. Bioinformatics Analysis

Kaplan–Meier survival analysis was conducted using the online KMplotter platform (https://kmplot.com/) to evaluate the prognostic significance of BECN1 and SLC7A11 expression in NSCLC patients receiving radiotherapy. The analysis was restricted to patients who underwent radiotherapy (*n* = 65). The JetSet best probe sets were selected for each gene (BECN1: 208946_s_at; SLC7A11: 209921_at), and the auto-selected optimal cutoff was used to stratify patients into high and low expression groups. Overall survival was plotted using Kaplan–Meier curves, and hazard ratios with 95% confidence intervals and log-rank P-values were calculated to determine statistical significance.

### 2.11. Statistical Analysis

All data are presented as mean ± SD from at least three independent experiments. Statistical significance was determined using one-way ANOVA followed by Tukey’s post hoc test for multiple comparisons. A *p*-value less than 0.05 was considered statistically significant.

## 3. Results

### 3.1. Cabozantinib Induces Ferroptosis and Suppresses STAT3/MCL1 Signaling in NSCLC Cells

To investigate the ferroptosis-modulating effects of cabozantinib in NSCLC, we selected lenvatinib and ripretinib as comparator VEGFR-targeting multi-kinase inhibitors with overlapping VEGFR inhibition but distinct secondary kinase profiles. Structural comparison revealed that these agents differ in their heterocyclic scaffolds and kinase-binding motifs (Table 1), suggesting that cabozantinib may exert unique intracellular effects beyond VEGFR blockade.

We first determine 24 h dose–response curves in H1299 and A549 cells for each TKI. Cabozantinib showed IC_50_ values of 11.65 µM in H1299 and 11.09 µM in A549, whereas lenvatinib and ripretinib caused only minimal cytotoxicity (Figure 1A). To assess ferroptosis involvement, cells were co-treated with TKIs (10 µM) and the ferroptosis inhibitors ferrostatin-1 (Fer-1, 5 µM) or deferoxamine (DFO, 200 µM), and erastin (20 µM) was included as a positive control. As expected, erastin-induced cell death was fully rescued by both Fer-1 and DFO, validating the ferroptosis-dependence of the assay (Figure 1B). Among the TKIs, only cabozantinib markedly reduced cell viability, and this reduction was significantly attenuated by co-treatment with either Fer-1 or DFO (*p* < 0.001), indicating a partial contribution of ferroptotic death to cabozantinib’s cytotoxic effect. By contrast, neither lenvatinib nor ripretinib at 10 µM elicited notable cell death, and their limited effects were unaffected by ferroptosis inhibition, suggesting minimal involvement of ferroptosis for these compounds under the same conditions.

To explore whether the differential cytotoxicity correlated with the modulation of pro-survival redox signaling, we examined the expression of STAT3 and its downstream effector MCL1, which have previously been implicated in the resistance to oxidative stress and suppression of ferroptosis. In H1299 and A549 cells, only cabozantinib suppressed phospho-STAT3 (Tyr705) and MCL1 in a dose-dependent manner (Figure 1C). Neither lenvatinib nor ripretinib altered STAT3 activation or MCL1 expression at any tested concentration. Crucially, the ferroptosis blockade did not impact STAT3 phosphorylation or MCL1 expression (Figure 1D), supporting the interpretation that cabozantinib reduces p-STAT3 via kinase-level inhibition rather than via downstream ferroptotic execution.

These findings suggest that the ability of cabozantinib to promote ferroptotic cytotoxicity in NSCLC cells is mechanistically linked to its suppression of the STAT3/MCL1 signaling axis, a property not shared by the other VEGFR TKIs tested.

### 3.2. Cabozantinib Induces Lipid Peroxidation and Ferroptotic Cell Death in NSCLC Cells

To evaluate whether ferroptosis underlies cabozantinib-induced cytotoxicity in NSCLC, we examined the accumulation of lipid peroxides, an essential biochemical hallmark of ferroptotic cell death. Flow cytometric analysis using the oxidation-sensitive probe C11-BODIPY 581/591 revealed a dose-dependent increase in lipid ROS in both H1299 and A549 cells treated with cabozantinib (Figure 2A). This signal shift from red to green fluorescence indicated enhanced lipid peroxidation following treatment.

Consistent with this, H1299 cells showed elevated levels of malondialdehyde (MDA), a terminal lipid peroxidation product, as detected by FITC-conjugated antibody staining at 48 h (Figure 2B). To verify whether these oxidative events were ferroptosis-related, we co-treated cells with ferroptosis inhibitors, Fer-1 and DFO. Both agents significantly suppressed cabozantinib-induced lipid ROS accumulation (Figure 2C), confirming the iron-dependent nature of this oxidative process.

In parallel, propidium iodide (PI) staining revealed an increase in membrane-compromised, PI-positive cells following cabozantinib treatment. This effect was similarly reversed by Fer-1 and DFO co-treatment (Figure 2D), indicating that lipid peroxidation led to regulated, non-apoptotic cell death characteristic of ferroptosis.

These findings demonstrate that cabozantinib triggers ferroptosis in NSCLC cells by promoting iron-dependent lipid peroxidation and compromising membrane integrity, two key hallmarks of ferroptotic death. This supports its role as a pharmacologic modulator of ferroptosis and highlights its potential in redox-targeted therapeutic strategies. 

### 3.3. Cabozantinib Enhances Radiation-Induced Suppression of Clonogenicity and Sphere Formation via Ferroptosis

To determine whether cabozantinib sensitizes NSCLC cells to ionizing radiation through ferroptosis, we performed clonogenic survival assays in H1299 and A549 cells. Cells were treated with a vehicle, radiation, cabozantinib, or the combination of radiation and cabozantinib, with or without the ferroptosis inhibitor Fer-1. Radiation or cabozantinib alone induced only modest reductions in clonogenic potential, whereas the combination led to a synergistic and pronounced suppression of colony formation (Figure 3A). This effect was partially reversed by Fer-1, indicating that ferroptosis contributes to the long-term loss of proliferative capacity induced by combined treatment.

To assess the impact of treatment on the self-renewal potential of tumor-initiating subpopulations, we conducted 3D sphere formation assays under similar treatment conditions. Radiation or cabozantinib alone moderately impaired spheroid generation in both NSCLC cell lines (Figure 3B). However, the combination treatment almost completely abolished sphere formation, and this inhibitory effect was partially alleviated by Fer-1, implicating ferroptosis in the suppression of stem-like properties.

Given the impaired spheroid generation upon combination treatment, we examined the canonical NSCLC stemness markers (e.g., SOX2, CD44, CD133, EpCAM) after irradiation with or without cabozantinib. Radiation alone increased all four markers at 24 h in both H1299 and A549. In contrast, co-treatment with cabozantinib attenuated these radiation-induced increases in a dose-dependent manner, consistent with diminished stem-like phenotypes (Figure 3C).

These data suggest that cabozantinib amplifies the radiosensitivity of NSCLC cells by triggering ferroptosis, thereby impairing both clonogenic survival and tumorsphere-forming capacity. This ferroptosis-mediated loss of regenerative potential may underlie the durable cytotoxic effects observed with cabozantinib–radiation combination therapy.

### 3.4. Cabozantinib Enhances Radiation-Induced Lipid Peroxidation and Ferroptosis

To mechanistically substantiate the involvement of ferroptosis in cabozantinib-mediated radiosensitization, we assessed key biochemical features of ferroptosis following radiation and cabozantinib co-treatment in NSCLC cells. Lipid ROS accumulation, measured by C11-BODIPY oxidation, was modestly increased by radiation or cabozantinib monotherapy. However, the combination resulted in a synergistic elevation of lipid peroxidation in both H1299 and A549cells (Figure 4A). This increase was partially suppressed by Fer-1, confirming that the observed lipid oxidation was ferroptosis-dependent.

Consistent with the lipid ROS findings, MDA levels were significantly higher in the combination group than in either monotherapy, and this increase was attenuated by Fer-1 (Figure 4B), supporting ferroptosis as the mode of redox-driven damage. Furthermore, propidium iodide (PI) staining revealed a marked increase in membrane-permeabilized, PI-positive cells following radiation and cabozantinib co-treatment (Figure 4C). This membrane damage, a hallmark of ferroptotic death, was again reversed by Fer-1, ruling out confounding necrotic or apoptotic mechanisms.

To determine whether redox imbalance and glutathione depletion were required for this ferroptotic response, we co-treated H1299 cells with radiation and cabozantinib in the presence of either 2-mercaptoethanol (2-ME) or reduced glutathione (GSH). Both antioxidants partially restored cell viability (Figure 4D), further implicating glutathione exhaustion and redox collapse as key drivers of ferroptotic vulnerability in this context.

These results collectively demonstrate that cabozantinib augments radiation-induced ferroptosis through iron-dependent lipid peroxidation, membrane destabilization, and glutathione depletion. This ferroptotic enhancement underpins its radiosensitizing effect and supports the rationale for combining ferroptosis-inducing agents with radiotherapy to overcome oxidative resistance in NSCLC.

### 3.5. Cabozantinib Enhances Radiation-Induced Ferroptosis via the MCL1/BECN1/SLC7A11 Axis

To dissect the upstream mechanism underlying cabozantinib-mediated ferroptotic radiosensitization, we examined the impact of treatment on glutathione homeostasis and cystine transport. In H1299 cells, neither radiation nor cabozantinib alone caused substantial changes in extracellular glutamate release or total intracellular GSH content. However, their combination significantly suppressed glutamate export and depleted GSH levels (Figure 5A), consistent with the inhibition of the system Xc^−^ transporter and collapse of cellular redox capacity.

To determine whether this functional inhibition was mediated via BECN1, a known suppressor of SLC7A11 activity, we performed the immunoprecipitation of endogenous BECN1 followed by immunoblotting for SLC7A11 and MCL1. Combined radiation and cabozantinib treatment markedly enhanced the BECN1–SLC7A11 interaction while concomitantly reducing MCL1 protein levels (Figure 5B). Given that MCL1 has been shown to sequester BECN1 and block its non-autophagic functions, these results suggest that MCL1 suppression by cabozantinib permits the release of BECN1 to inhibit SLC7A11 and promote ferroptosis.

To functionally validate this mechanism, we silenced BECN1 in A549 cells using siRNA and treated the cells with radiation and cabozantinib. Knockdown efficiency was confirmed by immunoblotting. Importantly, BECN1 silencing restored glutamate release and intracellular GSH levels to near-baseline levels despite combination treatment (Figure 5C), indicating the preservation of cystine uptake and antioxidant capacity. In parallel, the BECN1 knockdown attenuated the induction of lipid ROS (C11-BODIPY), MDA accumulation, and PI-positive cell death (Figure 5D,E), supporting its central role in executing ferroptosis. Furthermore, the stable knockdown of BECN1 via shRNA significantly rescued clonogenic survival in response to radiation and cabozantinib (Figure 5F), confirming that BECN1 is required not only for the biochemical initiation of ferroptosis, but also for its long-term cytotoxic consequences.

Together, these findings delineate a mechanistic pathway in which cabozantinib, via the suppression of MCL1, liberates BECN1 to engage and inhibit SLC7A11, resulting in glutathione depletion, lipid peroxidation, and ferroptotic cell death. This MCL1/BECN1/SLC7A11 axis provides a mechanistic basis for ferroptosis-driven radiosensitization and highlights a therapeutically actionable vulnerability in redox-adapted NSCLC cells.

### 3.6. Expression of BECN1 and SLC7A11 Correlates with Prognosis in NSCLC Patients Receiving Radiotherapy

To explore the translational relevance of the MCL1/BECN1/SLC7A11 ferroptosis axis in a clinical context, we analyzed public NSCLC survival datasets using the KMplotter platform (https://kmplot.com/), focusing on patients who underwent radiotherapy (n = 65). Probes were selected using the JetSet best match, and optimal expression thresholds were auto-determined by the algorithm.

The high expression of BECN1 (probe: 208946_s_at) was significantly associated with improved overall survival (hazard ratio [HR] = 0.43, 95% confidence interval [CI]: 0.22–0.82; log-rank *P* = 0.0092) (Figure 6A), consistent with its mechanistic role as a promoter of ferroptotic cell death in radiosensitized NSCLC. In contrast, elevated SLC7A11 expression (probe: 209921_at) was significantly correlated with poorer overall survival (HR = 2.43, 95% CI: 1.37–4.30; log-rank *P* = 0.0017) (Figure 6B), reflecting its function as a negative regulator of ferroptosis and contributor to redox-mediated radioresistance.

These clinical correlations mirror our in vitro findings, where BECN1 facilitated ferroptotic cell death through the direct inhibition of SLC7A11 following cabozantinib-mediated MCL1 suppression. Together, the data support a model (Figure 6C) wherein radiation and cabozantinib synergistically engage the MCL1/BECN1/SLC7A11 axis to overcome ferroptosis resistance in NSCLC cells. 

Importantly, BECN1 and SLC7A11 may serve not only as mechanistic mediators, but also as predictive biomarkers for radiotherapy responsiveness. Their expression levels could potentially stratify NSCLC patients for ferroptosis-targeted radiosensitization strategies. 

## 4. Discussion

Our findings identify a functional link between cabozantinib-induced radiosensitization and ferroptotic cell death in NSCLC cells, mediated through the modulation of the STAT3/MCL1/BECN1/SLC7A11 axis. Among the VEGFR-targeting TKIs tested, only cabozantinib induced ferroptotic features, characterized by elevated lipid peroxidation, loss of membrane integrity, and glutathione depletion, which were reversed by canonical ferroptosis inhibitors. These observations extend the pharmacological profile of cabozantinib beyond angiogenesis inhibition and suggest its capacity to destabilize redox buffering mechanisms under radiation-induced oxidative stress.

Mechanistically, we observed that cabozantinib uniquely downregulated phosphorylated and total STAT3, accompanied by a marked decrease in MCL1 protein levels. This effect was not replicated by lenvatinib or ripretinib, which do not target MET or AXL, indicating that inhibition of non-VEGFR kinases may be essential for the suppression of STAT3 signaling. The MET–STAT3 axis is well-established in NSCLC as a pro-survival and redox-protective circuit [17,18]. Thus, the ability of cabozantinib to collapse this axis provides a mechanistic rationale for its ferroptosis-sensitizing effect.

The downstream consequence of MCL1 downregulation was an increased association between BECN1 and SLC7A11, as shown by co-immunoprecipitation, indicating that the loss of MCL1 relieves BECN1 from sequestration. This is consistent with previous studies in other cancer models, where MCL1 was shown to limit ferroptosis by directly binding and inactivating BECN1 [13,14]. Functionally, the partial knockdown of BECN1, consistent with its essential role in cellular homeostasis, was nonetheless sufficient to attenuate the ferroptotic phenotype induced by cabozantinib and radiation. Even with incomplete depletion, BECN1 silencing restored intracellular glutathione, suppressed lipid peroxidation, and rescued clonogenic survival, thereby underscoring the functional relevance of BECN1 in this pathway. Collectively, these results support a model in which BECN1 acts as a key effector, coupling upstream redox-regulatory changes to the execution of ferroptosis.

The observation that cabozantinib, but not other structurally related VEGFR TKIs, engages the MCL1/BECN1/SLC7A11 pathway suggests this effect is not a class-wide phenomenon but rather a compound-specific property. This implies that the inhibition of MET or AXL–STAT3 signaling may be necessary for ferroptosis induction in this context. More importantly, we noted that radiation alone increased p-STAT3, consistent with prior reports that irradiation can trigger compensatory STAT3 activation via IL-6/JAK signaling and DNA damage–NF-κB crosstalk [19,20,21]. Cabozantinib partially suppressed this radiation-induced p-STAT3, in line with its inhibition of upstream MET/AXL–STAT3 signaling. Thus, the ability of cabozantinib to dampen STAT3 activation may be a prerequisite for ferroptosis induction in this context. From a therapeutic standpoint, tumors with persistent STAT3 activation, a hallmark of therapy-resistant NSCLC, may represent a subset particularly amenable to ferroptosis-based radiosensitization. The ability of cabozantinib to simultaneously inhibit angiogenic signaling and redox resistance could thus provide a therapeutic advantage over regimens that rely on synthetic ferroptosis inducers, which are limited by safety margins and clinical availability.

Nevertheless, our study has several limitations. The absence of in vivo validation precludes conclusions about therapeutic efficacy in complex tumor environments. Additionally, while our results support a causal relationship between cabozantinib, STAT3/MCL1 suppression, and ferroptosis, upstream kinase dependencies (e.g., MET vs AXL vs RET) were not dissected. Kinase-selective inhibition or CRISPR-based functional mapping could help clarify these contributions. Furthermore, the interaction between ferroptosis and immune modulation under radiotherapy remains an important open question, particularly given the known immunomodulatory effects of cabozantinib in other settings.

## 5. Conclusions

This study provides mechanistic evidence that cabozantinib enhances radiation-induced ferroptosis in NSCLC cells through a pathway involving the suppression of STAT3/MCL1, activation of BECN1, and inhibition of SLC7A11. These findings reveal a non-canonical role for cabozantinib in redox vulnerability modulation and suggest that ferroptosis can be leveraged as a radiosensitization strategy in tumors with intact BECN1 function and reliance on MCL1-driven redox protection. Future studies integrating in vivo modeling and biomarker-guided patient stratification will be crucial for clinical translation.

## Figures and Tables

**Figure 1 cancers-17-02950-f001:**
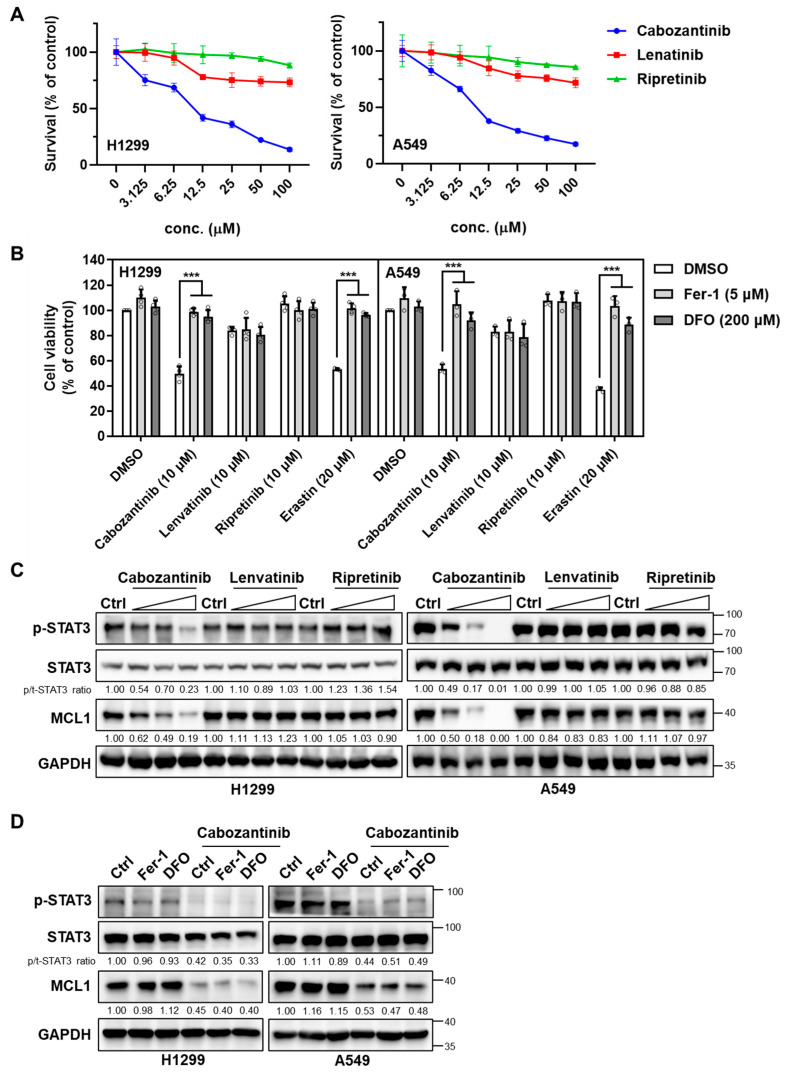
Cabozantinib uniquely triggers ferroptosis-linked cytotoxicity and suppresses the STAT3–MCL1 axis in NSCLC cells. (**A**) Dose–response curves in H1299 and A549 cells treated for 24 h with cabozantinib, lenvatinib, or ripretinib across a six-point, twofold serial dilution (3.125–100 µM). Viability was normalized to vehicle (0.1% DMSO) and plotted as mean ± SD (*n* = 3 independent experiments). (**B**) Ferroptosis-rescue assay in H1299 and A549 cells treated with cabozantinib, lenvatinib, or ripretinib (10 μM each), or the ferroptosis inducer erastin (20 μM), with or without Fer-1 (5 μM) or DFO (200 μM) for 24 h. Data are presented as mean ± SD from three independent biological replicates. ***, *p* < 0.001. (**C**) Immunoblotting analysis of total STAT3, phospho-STAT3 (Tyr705), and MCL1 in H1299 and A549 cells treated with TKIs (3.125, 6.25, and 12.5 μM for 24 h). (**D**) H1299 and A549 cells were treated with or without 10μM cabozantinib at the indicated ferroptosis inhibitors (5 μM Fer-1 or 200 μM DFO) for 24 h and immunoblotted as in (**C**).

**Figure 2 cancers-17-02950-f002:**
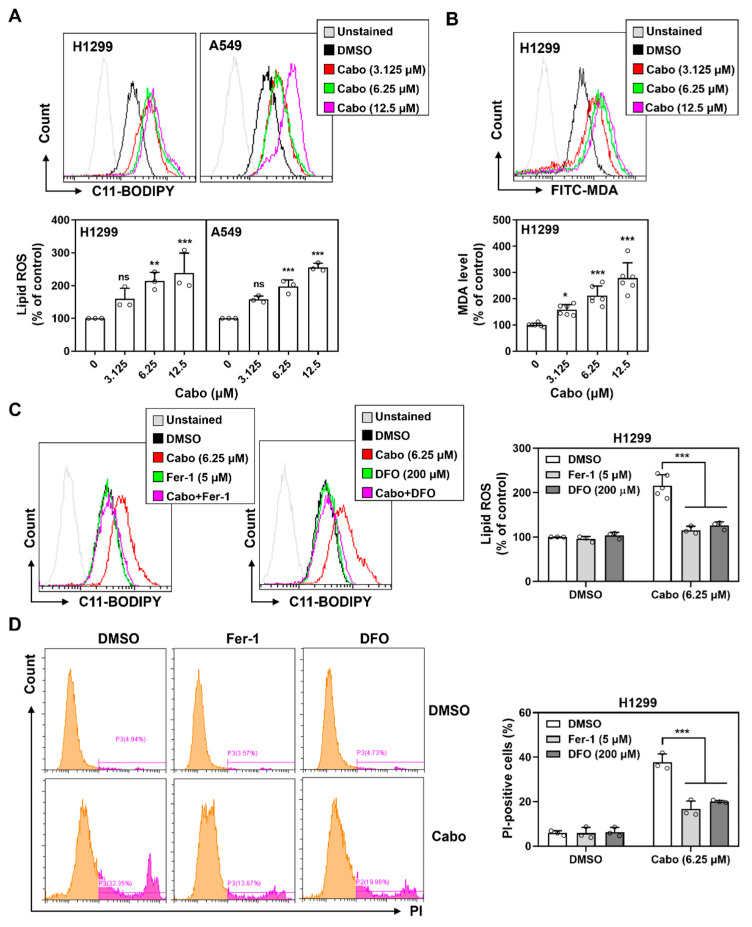
Cabozantinib induces ferroptosis via iron-dependent lipid peroxidation. (**A**) Flow cytometric analysis of lipid ROS using C11-BODIPY 581/591 in H1299 and A549 cells treated with cabozantinib (Cabo, 3.125, 6.25, or 12.5 μM) or DMSO for 24 h. (**B**) H1299 cells were stained with FITC-conjugated MDA antibody following 48 h treatment with cabozantinib at the indicated doses. (**C**) Lipid ROS levels at 24 h in H1299 cells treated with DMSO, 6.25 μM Cabo, 5 μM Fer-1, 200 μM DFO, or Cabo combined with Fer-1 or DFO. (**D**) Flow cytometry of PI-positive cells at 48 h under the same treatment conditions as (**C**). Data are presented from at least three independent experiments. *, *p* < 0.05; **, *p* < 0.01; *** *p* < 0.001. The orange peak denotes PI-negative viable cells, while the pink peak represents PI-positive non-viable/dead cells.

**Figure 3 cancers-17-02950-f003:**
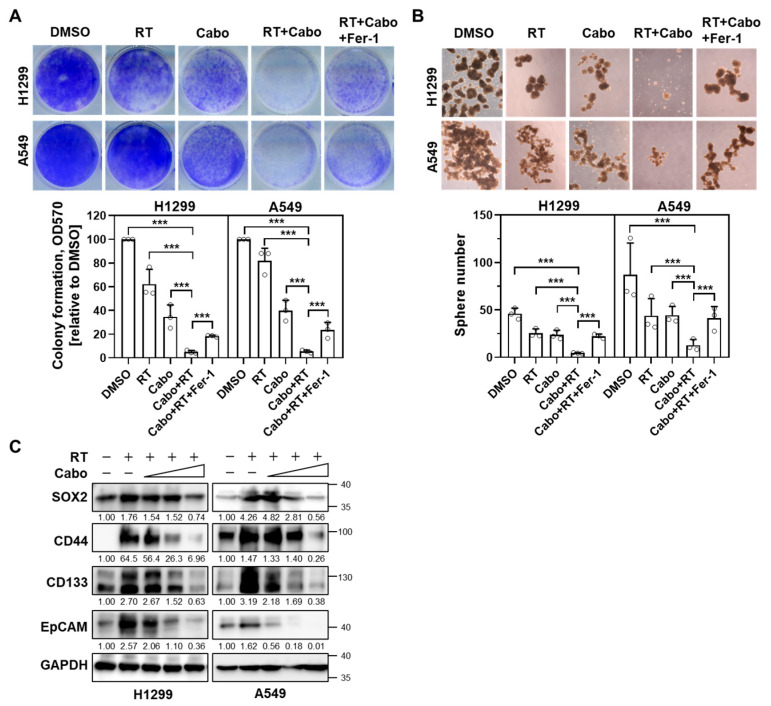
Cabozantinib enhances radiation-induced suppression of clonogenicity and sphere formation via ferroptosis. (**A**) Clonogenic survival assays of H1299 and A549 cells treated with DMSO, 4 Gy radiation (RT), 6.25 μM Cabo, or a combination of RT and Cabo, with or without 5 μM Fer-1. (**B**) Sphere formation assay in H1299 and A549 cells treated with DMSO, 4 Gy RT, 3.125 μM Cabo, or a combination of RT and Cabo, with or without 5 μM Fer-1. Data are presented for three independent experiments. *** *p* < 0.001. (**C**) H1299 and A549 cells were irradiated and treated without or with cabozantinib at the indicated concentrations (3.125, 6.25, 12.5 µM) for 24 h. Protein levels of SOX2, CD44, CD133, and EpCAM were analyzed by immunoblotting.

**Figure 4 cancers-17-02950-f004:**
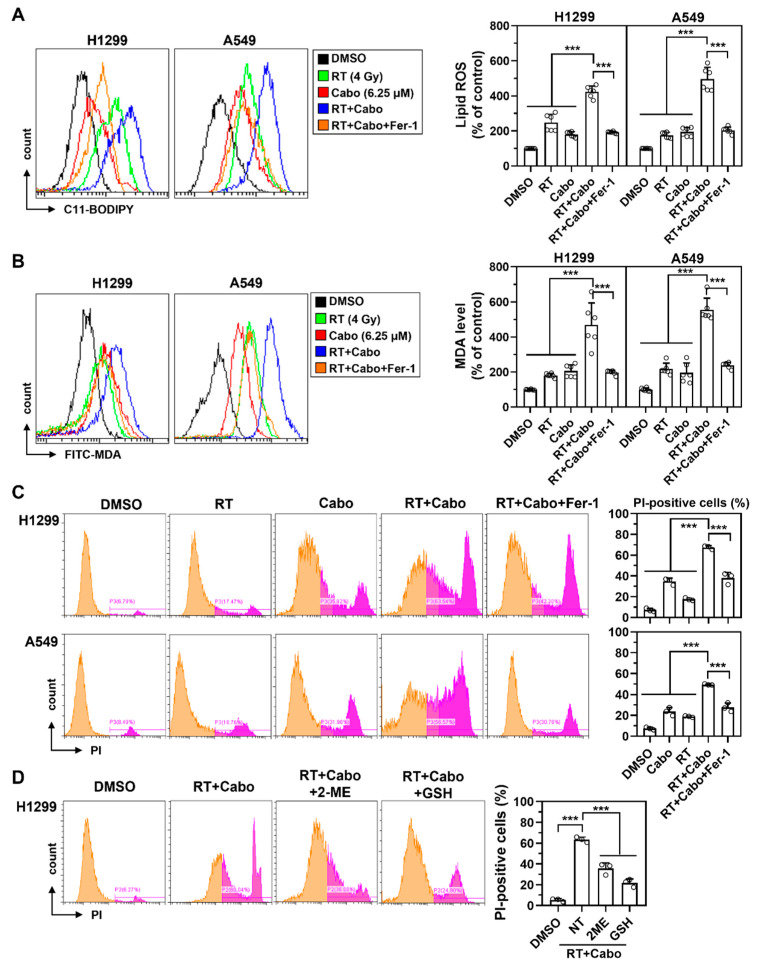
Cabozantinib enhances radiation-induced lipid peroxidation and ferroptotic death in NSCLC cells. (**A**–**C**) Flow cytometry analysis of C11-BODIPY oxidation (lipid ROS) at 24 h (**A**), MDA levels at 48 h (**B**), and PI-positive cells at 48 h (**C**) in H1299 and A549 cells treated with DMSO, 4 Gy RT, 6.25 μM Cabo, RT combined with Cabo, or RT and Cabo with 5 μM Fer-1. (**D**) H1299 cells were treated with radiation and cabozantinib in the presence of 2-ME or reduced GSH. Data are presented for at least three independent experiments. *** *p* < 0.001. The orange peak denotes PI-negative viable cells, while the pink peak represents PI-positive non-viable/dead cells.

**Figure 5 cancers-17-02950-f005:**
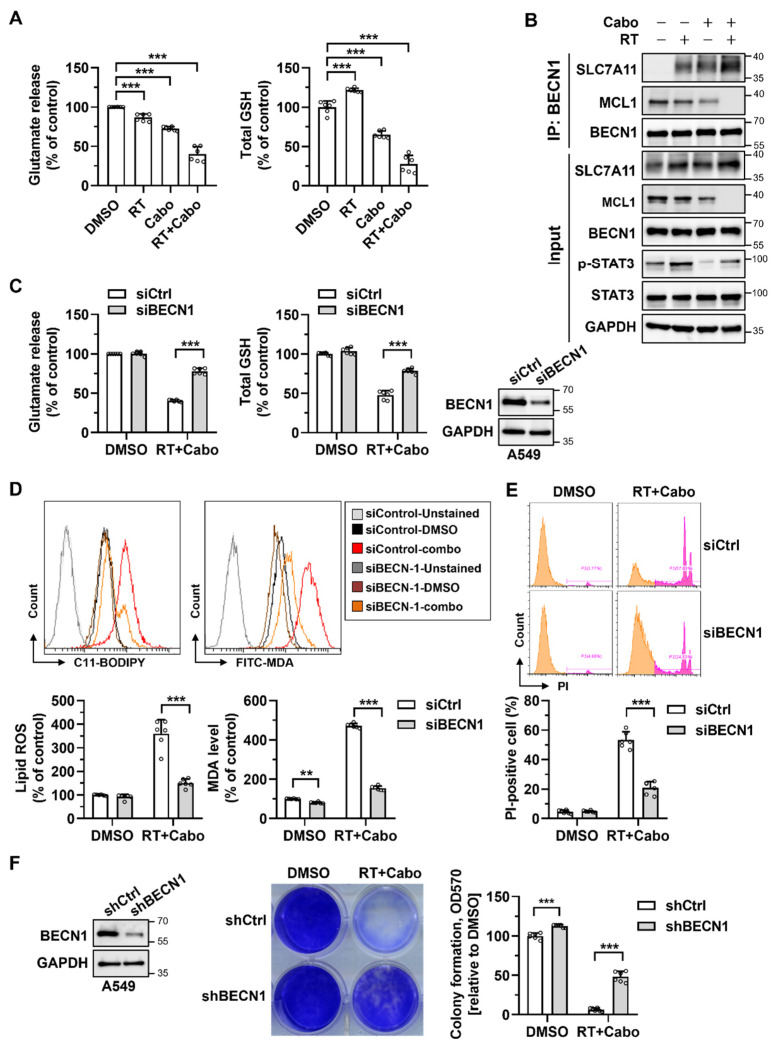
Cabozantinib promotes ferroptosis through the MCL1/BECN1/SLC7A11 pathway. (**A**) Glutamate release and total GSH levels in H1299 cells treated with DMSO, 4 Gy RT, 6.25 μM Cabo, or the combination of RT and Cabo. (**B**) Immunoprecipitation (IP) of BECN1 in cells under the same conditions as (**A**), followed by immunoblotting for SLC7A11, MCL1, and BECN1. (**C**) A549 cells were transfected with control siRNA (siCtrl) or BECN1 siRNA (siBECN1), treated with DMSO or the combination of radiation and cabozantinib. Immunoblotting analysis confirms the BECN1 knockdown; meanwhile, glutamate release and GSH levels were measured. (**D**,**E**) Lipid ROS (C11-BODIPY) and MDA levels (**D**) and PI-positive cells (**E**) were measured under the same conditions as (**C**). (**F**) Clonogenic survival of A549 cells expressing shCtrl or shBECN1 treated with DMSO or the combination of radiation and cabozantinib. Data are presented for at least three independent experiments. *** *p* < 0.001. The orange peak denotes PI-negative viable cells, while the pink peak represents PI-positive non-viable/dead cells.

**Figure 6 cancers-17-02950-f006:**
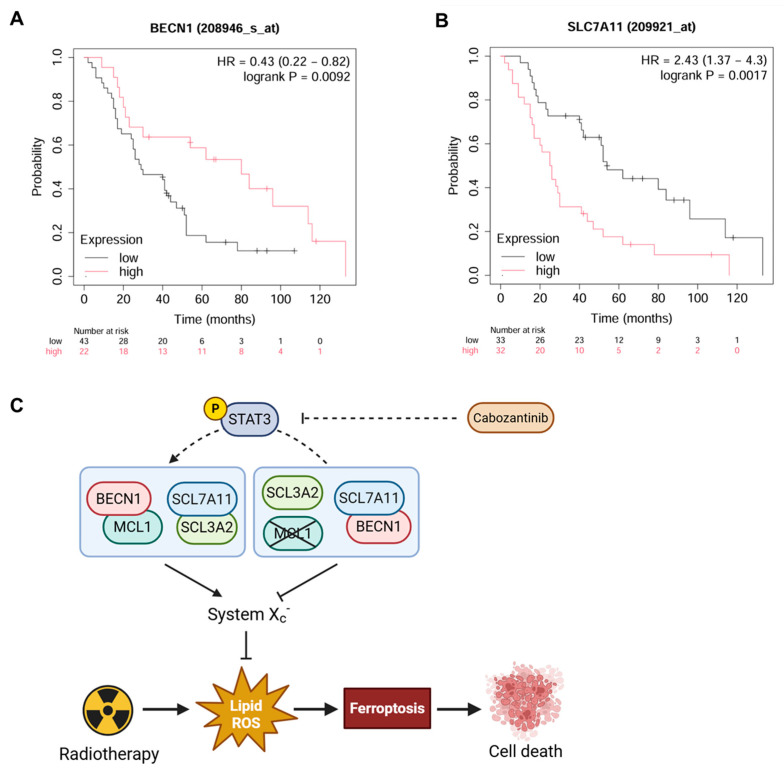
Prognostic significance of BECN1 and SLC7A11 expression in radiotherapy-treated NSCLC patients. Kaplan–Meier survival curves of NSCLC patients (n = 65) who received radiotherapy stratified by BECN1 (**A**) and SLC7A11 (**B**) expression levels. The hazard ratios with 95% confidence intervals and log-rank *P*-values are indicated. (**C**) Proposed model of the MCL1/BECN1/SLC7A11 axis regulating ferroptosis in response to radiotherapy in NSCLC.

**Table 1 cancers-17-02950-t001:** Structures and target spectra of multi-kinase inhibitors used in this study.

Inhibitor	Cabozantinib	Lenvatinib	Ripretinib
Structure	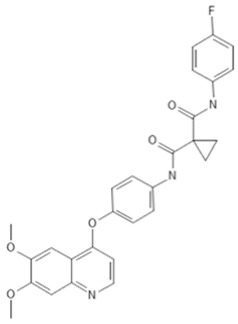	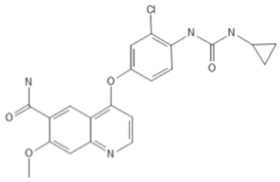	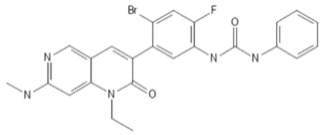
Primary Approved Targets	VEGFR1–3, MET, AXL	VEGFR1–3, FGFR1–4, PDGFRα, RET, KIT	KIT, PDGFRA
Additional Kinases (reported activity)	RET, KIT, FLT3, ROS1, TYRO3, MER	–	VEGFR2, PDGFRB, TIE2, BRAF

## Data Availability

The data used and analyzed during the current study are available from the corresponding authors on reasonable request.

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
