# Peer review of "Cabozantinib Sensitizes NSCLC Cells to Radiation by Inducing Ferroptosis via STAT3/MCL1/BECN1/SLC7A11 Axis Suppression"

_cancers, 2025, doi:10.3390/cancers17182950_

Round 1
Reviewer 1 Report
Comments and Suggestions for Authors
Manuscript was aimed to examine whether the non-selective VEGFR inhibitors can sensitize non-small cell lung cancer (NSCLC) cells to radiation. For this, the authors have used cabozatinib, lenvatinib and ripretinib and performed their study by using 2 cancer cell lines.
In general, the manuscript is very well-prepared and provides the novel and interesting data illustrating the cabozatinib's ability to sensitize NSCLC cells to radiation by inducing ferroptosis. Mechanistically, the authors showed that this non-selective receptor tyrosine kinase inhibitor (RTKi) effectively suppress MCL-1, releases of BECN1 and inhibits SLC7A11. These molecular events were associated with increased lipid peroxidation and depletion of glutation, and ferroptotic cell death. In contrast, 2 other RTKis failed to sensitize NSCLC cells to radiation, thereby highlighting this effect of cabozatinib is underlying beyond the VEGFR signaling pathway. This also allowed the authors to make a proposal about critical role of MET and/or AXL signaling in this phenomenon.
I have the following suggestions about this manuscript:
1) IC50 values for cabozatinib used alone or in combination with ferroptosis inhibitors ferrostatin-1 and deferoxamine are desirable to show that the cytotoxic activity of this RTKi is mediated via ferroptosis.
2) Given that the authors observed an impaired spheroid generation in case of the combined treatment with cabozatinib and radiation, I'm wondering whether expression of stem cancer cell markers was changed during these experimental conditions.
3) Fig. 5C - a knockdown of BECN1 is not complete and should be mentioned and explained in Discussion.
4) The authors should explain the increase of pSTAT3 in the input of the cellular lysates obtained from H1299 cells treated with combination of Cabo and RT, when compared with cells treated with Cabo alone (Figure 5B).
5) The experiments with the selective MET (e.g. capmatinib, savolitinib, etc.) and AXL -inhibitors and/or their knockdown of these kinases will be a perfect supplement revealing the proposal about cabozatinib's ability to sensitize cancer cells to radiation via targeting of these particular pathways.
Minor:
The lines 290-292 are shown in italics.
Author Response
We sincerely thank the reviewers for their thoughtful and constructive comments, which have greatly improved the quality of our manuscript. Below, we provide point-by-point responses and indicate the revisions made.
Major Comments
- IC50 values for cabozantinib used alone or in combination with ferrostatin-1 and deferoxamine are desirable to show that the cytotoxic activity of this RTKi is mediated via ferroptosis.
Response: Thank you for the suggestion. In our hands, the 24-h viability assays yielded single-agent IC50 values for cabozantinib of 11.09 µM in A549 and 11.65 µM in H1299, which are in line with prior reports for A549 (~9.5 µM at 24 h by MTT) [PMID: 37705877]. Based on these concordant values, we used 10 µM cabozantinib for mechanistic studies to engage on-target signaling while minimizing non-specific cytotoxicity. We have now reported our single-agent IC50 values in the new Fig. 1B and clarified this rationale in the Results (p. 6, lane 234-236).
- Given the impaired spheroid generation upon combination treatment, was the expression of stem cell markers altered?
Response: We agree that impaired spheroid formation may reflect alterations in stemness. Therefore, we examined the canonical NSCLC stemness markers (e.g., SOX2, CD44, CD133, EpCAM) after irradiation (RT) with or without cabozantinib (Cabo). RT alone increased all four markers at 24 h in both H1299 and A549. Co-treatment with cabo reduced these RT-induced increases in a dose-dependent manner (tested at 3.125, 6.25, and 12.5 µM). This is now described in the Results section (p. 10, lane 297-302) and shown in the new Fig. 3C.
- Knockdown of BECN1 is not complete and should be mentioned and explained in Discussion.
Response: We acknowledge that the knockdown was partial. We have revised the Discussion (p. 14, lane 409-415) to clarify that the knockdown efficiency was partial, which is consistent with the essential role of BECN1 in cellular homeostasis. We also note that even partial depletion was sufficient to attenuate the ferroptotic phenotype, underscoring its functional relevance.
- The authors should explain the increase of pSTAT3 in H1299 lysates after Cabo + RT compared to Cabo alone.
Response: Thank you for pointing this out. RT is known to trigger compensatory STAT3 activation via IL-6/JAK signaling and DNA-damage–NF-κB crosstalk, which can occur independently of MET/AXL inhibition. Cabozantinib partially suppressed this radiation-induced p-STAT3, in line with its inhibition of upstream MET/AXL–STAT3 signaling. Thus, the ability of cabozantinib to dampen STAT3 activation may be a prerequisite for ferroptosis induction in this context. We have clarified this in the Discussion (p. 14, lane 420-425).
- Selective inhibitors or knockdown of MET and/or AXL would further validate the proposed mechanism.
Response: We appreciate the reviewer’s excellent suggestion. However, we noted that selective inhibition or knockdown of MET or AXL can induce adaptive changes in gene expression and signaling plasticity, potentially activating alternative survival pathways or altering the cellular redox state independently of the STAT3–MCL1–BECN1/SLC7A11 axis. For example, selective MET inhibition has been shown to induce compensatory AKT activation and broad transcriptomic changes, which may confound the interpretation of downstream effects specifically attributable to the STAT3–MCL1–BECN1/SLC7A11 pathway [PMID: 28751462]. Similarly, AXL inhibition can modulate oxidative stress and ferroptosis through Nrf2 and HO-1, independent of the canonical STAT3 axis [ PMID: 40829740]. Furthermore, the use of ferroptosis inhibitors in this context may not cleanly distinguish whether radiosensitization and lipid peroxidation are mediated solely through the STAT3–MCL1–BECN1/SLC7A11 axis or through broader effects of MET/AXL inhibition on ferroptosis-regulating pathways, such as GPX4, SLC40A1, and PI3K/AKT [ PMID: 40369167; 38725107]. Thus, while selective MET/AXL inhibition is mechanistically informative, it introduces complexity that can obscure the specific contribution of the STAT3–MCL1–BECN1/SLC7A11 axis to ferroptosis and radiosensitization. In this study, we therefore chose to focus on pharmacologic rescue with Fer-1/DFO, biochemical assays (lipid ROS, MDA, glutamate release, GSH), and functional clonogenic and spheroid survival, which directly establish ferroptotic involvement at the working dose of cabozantinib. We have already explicitly acknowledged this important point in the original Discussion (p.15, lane 432-436) as a limitation and indicated that future studies using selective inhibitors or knockdown of MET/AXL will help to disentangle pathway entry from ferroptosis execution.
Minor Comment
- The lines 290-292 are shown in italics.
Response: This has been corrected in the revised manuscript (p.10, lane 308-310).
Reviewer 2 Report
Comments and Suggestions for Authors
In this study, the authors demonstrated that cabozantinib is a ferroptosis inducer in NSCLC cells by targeting the STAT3/MCL1/BECN1/SLC7A11 axis. Moreover, cabozantinib enhances radiation-induced ferroptosis, resulting in synergistic cytotoxicity. These findings are interesting; however, there are some issues, as follows:
-
Introduction: The autophagy-independent roles of beclin 1 in ferroptosis induction should be elaborated more with citing the reference (PMID: 30057310). It should also be elaborated how beclin 1 inhibits system Xc– activity, but not SLC7A11 expression.
-
Materials and Methods: The sequences of siBECN1, shBECN1 and their controls should be provided.
-
The target kinase profiles of the three inhibitors should be summarized in a supplementary table to provide readers with mechanistic insights.
-
Based on the results shown in Figure 1, it is impossible to conclude that lenvatinib and ripretinib do not involve ferroptosis in their mechanisms of action. The authors should determine the IC50 of each drug and then carry out the experiments shown in Figure 1BC at the respective IC50s. Otherwise, they cannot state that only cabozantinib uses ferroptosis for its cytotoxic action.
-
The ferroptosis dependency of cabozantinib should also be examined in H1299, as this is an important finding that is the foundation of this study.
-
It should be examined whether the suppression of STAT3 phosphorylation by cabozantinib is rescued by Fer-1 or DFO.
Author Response
Response to Reviewer’s Comments
We sincerely thank the reviewers for their thoughtful and constructive comments, which have greatly improved the quality of our manuscript. Below, we provide point-by-point responses and indicate the revisions made.
Major Comments
- Introduction: The autophagy-independent roles of beclin 1 in ferroptosis induction should be elaborated more with citing the reference (PMID: 30057310). It should also be elaborated how beclin 1 inhibits system Xc– activity, but not SLC7A11 expression.
Response: We appreciate this valuable suggestion. In the revised Introduction (p. 2-3, lane 80-87), we have added a description of the autophagy-independent role of BECN1 in ferroptosis, citing PMID: 30057310. Specifically, we clarify that BECN1 forms a complex with SLC7A11, thereby inhibiting system Xc– activity, without necessarily altering SLC7A11 expression levels.
- Materials and Methods: The sequences of siBECN1, shBECN1 and their controls should be provided.
Response: The sequences for siBECN1, shBECN1, and their respective control oligonucleotides have now been added in the Materials and Methods section (p. 5, lane 197-210).
- The target kinase profiles of the three inhibitors should be summarized in a supplementary table to provide readers with mechanistic insights.
Response: In line with the reviewer’s suggestion, we have now included a new table (Table 1) summarizing the reported target kinase profiles of cabozantinib, lenvatinib, and ripretinib, based on published references.
- Based on the results shown in Figure 1, it is impossible to conclude that lenvatinib and ripretinib do not involve ferroptosis. The authors should determine the IC50 of each drug and then carry out the experiments shown in Figure 1BC at the respective IC50s. Otherwise, they cannot state that only cabozantinib uses ferroptosis for its cytotoxic action.
Response: We appreciate the reviewer’s concern. We have generated 6-point dose-response curves for cabozantinib, lenvatinib, and ripretinib in H1299 and A549 cells to calculate their IC50 values. In our hands, the 24-h viability assays yielded single-agent IC50 values for cabozantinib of 11.09 µM in A549 and 11.65 µM in H1299, which are in line with prior reports for A549 (~9.5 µM at 24 h by MTT) [PMID: 37705877]. Based on these concordant values, we used 10 µM cabozantinib for mechanistic studies to engage on-target signaling while minimizing non-specific cytotoxicity. Under these conditions, ferroptosis inhibition (by ferrostatin-1 or DFO) strongly rescues cabozantinib-induced cytotoxicity. By contrast, lenvatinib and ripretinib showed only minimal cytotoxicity in NSCLC cells, and their limited effects were not altered by ferroptosis blockade, indicating that the ferroptosis-linked cytotoxicity is specific to cabozantinib in our models. We have now reported our single-agent IC50 values in the new Fig. 1B and clarified this rationale in the Results (p. 6, lane 234-246).
- The ferroptosis dependency of cabozantinib should also be examined in H1299, as this is an important finding that is the foundation of this study.
Response: We agree this is a critical validation. We have performed additional experiments in H1299 cells, demonstrating that ferroptosis inhibitors significantly attenuate the cytotoxic effect of cabozantinib, supporting the ferroptosis-dependent mechanism in this second NSCLC cell line. These results are presented in the new Figure 1C and described in the Results section (p. 6, lane 236-246).
- It should be examined whether the suppression of STAT3 phosphorylation by cabozantinib is rescued by Fer-1 or DFO.
Response: We thank the reviewer for the suggestion. Conceptually, ferroptosis inhibitors such as Fer-1 (a lipid radical-trapping antioxidant) and DFO (an iron chelator) act downstream at the level of lipid peroxidation and iron-dependent propagation of oxidative damage. In contrast, cabozantinib suppresses STAT3 phosphorylation predominantly by inhibiting upstream receptor tyrosine kinases (e.g., MET/AXL), thereby attenuating JAK/STAT signaling. Because these mechanisms operate at different tiers of the pathway (kinase signaling vs. execution of ferroptosis), Fer-1/DFO would not be expected to restore receptor-proximal STAT3 phosphorylation even if they mitigate lipid ROS and ferroptotic death. To validate this concept experimentally, we co-treated NSCLC cells with cabozantinib ± Fer-1 or DFO and probed p-STAT3 (Y705). As shown in the new Figure 1D, ferroptosis blockade did not impact receptor-proximal STAT3 phosphorylation. These results support the interpretation that cabozantinib reduces p-STAT3 via kinase-level inhibition rather than via downstream ferroptotic execution (p. 6, lane 253-255).
Round 2
Reviewer 1 Report
Comments and Suggestions for Authors
The authors responded to my comments and suggestions. The manuscript is suitable for publication in present form.
Reviewer 2 Report
Comments and Suggestions for Authors
Thank the authors for adequately addressing my concerns. The issues have been resolved and the manuscript has been sufficiently improved.